# The Winding Road to Equal Care: Attitudes and Experiences of Prescribing ADHD Medication among Pediatric Psychiatrists: A Qualitative Study

**DOI:** 10.3390/ijerph20010221

**Published:** 2022-12-23

**Authors:** David Sjöström, Olof Rask, Linda Welin, Marie Galbe Petersson, Peik Gustafsson, Kajsa Landgren, Sophia Eberhard

**Affiliations:** 1Office for Psychiatry, Habilitation and Aid, Child and Adolescent Mental Health Services, Region Skåne, 20502 Malmö, Sweden; 2Department of Clinical Sciences Lund, Child and Adolescent Psychiatry, Lund University, 22100 Lund, Sweden; 3Department of Health Sciences, Faculty of Medicine, Lund University, 22240 Lund, Sweden; 4Office for Psychiatry and Habilitation, Psychiatry Research Skåne, Region Skåne, 22185 Lund, Sweden

**Keywords:** ADHD, prescription variation, child and adolescent psychiatry, qualitative method, parents

## Abstract

Evidence is lacking on how to understand the reasons for variations, both in prevalence of ADHD and ADHD medication prescribing patterns in children and adolescents, within Region Skåne. These variations are not in line with current national clinical guidelines and seem to have increased over time. This qualitative interview study illuminates pediatric psychiatrists’ attitudes toward ADHD and their experiences of prescribing ADHD medication. Eleven pediatric psychiatrists described the complex interplay of variables that they experienced while assessing a child, which had influence on their decision to prescribe medication. Being part of a local unit’s culture influenced how ADHD medications were prescribed. They wished that the assessment of the child’s symptoms was consistent with guidelines in every unit but noted that such alignment was not implemented. They pointed out that an ADHD diagnosis is dependent on the surrounding’s motivation and capacity to adapt to the present state of the child. The participants described how they balanced clinical guidelines with demands from the family, as well as from society at large. Their personal attitudes and clinical experiences towards diagnosing and prescribing medications to children with ADHD influenced their decisions. The study adds information about how attitudes may lead to variation in diagnostics and therapy.

## 1. Introduction

Attention deficit hyperactivity disorder (ADHD) affects about 5–6 percent of all Swedish children and is characterized by impaired attention, impulsiveness, and hyperactivity [1]. These symptoms can cause a wide range of difficulties both in school, at home and during social activities. ADHD is a significant risk factor for reduced school performance, unemployment, substance abuse, mental illness, co-morbidity and mortality [2,3,4,5,6,7]. Worldwide, the prevalence of ADHD diagnosis in children varies between 3.4–7.2% [1,8] and the prevalence for prescribed ADHD medication between 0.27–6.69% [9]. Given the potential extensive problems that ADHD may cause children, their families and society, it is of importance to identify those with disabling symptoms and offer evidence-based interventions. Studies indicate that both pharmacological and psychological interventions positively influence the core symptoms of hyperactivity and impaired attention in ADHD and increase daily function [10].

Over the last decades, the numbers of children and families seeking psychiatric health care have increased substantially in many countries [11,12] and this increase is also apparent in pediatric psychiatry in Region Skåne. It can be substantially explained by a corresponding increase in children diagnosed with ADHD [13]. Despite robust evidence on ADHD, the diagnosis is perceived by some as controversial, particularly with regard to the topic of prescribing medication or not [14]. These major changes in incidence have become subject of rising public attention that might affect professionals in child and adolescent mental health services (CAMHS). Almost every day, pediatric psychiatric issues are highlighted in the Swedish daily media, sometimes with strong opinions questioning existing routines for diagnosis and treatment: Should more children be enrolled in psychiatric clinics? Less? Studies have indicated increased stigma around being diagnosed with ADHD [15]. There is also discussion whether ADHD is a chronic condition or not, with a recent meta-analysis demonstrating that just 15–65% of children diagnosed with ADHD met diagnostic criteria for ADHD at age 25 [16].

Preliminary results highlighted significant differences in the prevalence rates of children with ADHD between the four pediatric psychiatric clinics in Region Skåne, with some clinics in line with national numbers whilst others are far below. The preliminary results also indicated that for those diagnosed with ADHD, less ADHD medication was prescribed per patient at the clinics with low ADHD prevalence. These differences indicate a variation in adherence to current guidelines and seem to increase over time (unpublished clinical data). Previous research suggests that several reasons might add to this, including the prescribing physicians’ attitudes [17]. These differences are not unique for Region Skåne or Sweden; a significant variation of prescribing patterns has been demonstrated in Norway [17], Denmark [18,19], the US [20,21] and Germany [22] as well.

It is essential to find out what the variations in diagnosing and prescribing patterns are motivated by in order to promote equal health-care utilization. The aim of this study was to investigate the attitudes and experiences of diagnosing and prescribing medication for ADHD among senior consultants in pediatric psychiatry.

## 2. Materials and Methods

### 2.1. Design

The study is based on a qualitative design with semi-structured interviews analyzed with interpretative content analysis on a latent level [23]. This structured method reveals depth and meaning in the informants’ narratives and is therefore well suited for investigating attitudes and experiences.

### 2.2. Setting

The study was conducted at CAMHS in Region Skåne in southern Sweden. CAMHS Skåne comprises of four clinics and covers a catchment area of about 300,000 children and adolescents up to the age of 18 years. CAMHS Skåne consists of 23 outpatient units, with four tertiary out-patient units for children with psychosis, bipolarity, post-traumatic stress disorder, eating disorder and gender dysphoria. The four large cities in the region have day care units. The only emergency unit is located in Malmö and consists of a pediatric psychiatric emergency room and two inpatient units with 21 beds in total.

### 2.3. Participants

All 36 senior consultants at the CAMHS in Region Skåne, excepting for those involved in this study, were invited to take part. Thirteen consultants agreed to take part, all of them employed at outpatient units, and eleven interviews were conducted. Five of the participating consultants were men and six women, with specialist experience of between two and 17 years (mean seven years). All of them frequently met children with ADHD in their daily practice.

### 2.4. Data Collection

All interviews were performed by DS between September and December 2021. Due to the coronavirus pandemic, the interviews were conducted online via Zoom. The interview length varied between 33 and 48 min (median 38 min). The interviews were recorded with a voice recorder. The interviewer gave open questions based on the variations in assessment and prescription pattern within the region, following a semi-structured interview guide with follow-up probing questions. At the end, the participants were asked if they had any final comments.

### 2.5. Data Analysis

The recorded interviews were transcribed verbatim and analyzed by latent content analysis [23] by DS and KL. As a first step, the interviews were read. Then, the transcripts were divided into meaning units, which were condensed and labelled with a descriptive code close to the original text, with a low level of abstraction. The next step when interrelated codes were categorized into sub-themes and themes involved an interpretation of the informants´ narratives, looking for the underlying meaning. Striving to make the participants’ voices heard, the process involved a de-textualization and re-textualization when formulating the sub-themes and themes, and a movement between the parts and the whole [23]. The result was presented to the other authors, initiating a collective reflection resulting in a formulation of Discussion and Conclusion.

### 2.6. Ethical Considerations

The study was approved by the Swedish Ethical Review Authority (Ethical Review Board no. 2021-01563). After the participants had read the study information, they verbally agreed to the informed consent in the voice recording. All participants were to some extent colleagues with the interviewer DS, although none were mentors, supervisors or in other positions of potential hierarchical power. Only DS and KL read the interviews, and only DS knew who participated.

## 3. Results

The analysis revealed three themes with underlying sub-themes (Table 1) illuminating attitudes of senior consultants in pediatric psychiatry towards ADHD diagnosis and medication. The result is portrayed by quotes.

### 3.1. Meeting the Family

The first theme illuminates the complex interplay of variables that the clinicians experienced, with those present with them in the room, while assessing a child. These factors were described as influencing their decision to prescribe medication on the same day, to wait a while or not prescribe at all.

#### 3.1.1. Defining Normal Function

The clinicians discussed the fundamental challenge of defining normal function, how it is connected to a child’s maturity over time and depending on the surrounding circumstances. They acknowledged that their own attitude around normality influenced how they as clinicians weighed a prescription of ADHD medication versus trying other interventions. They argued that the predominant culture, to some extent, defines normal function and whether the criterion for a diagnosis is met, along with how those definitions have changed over the last decades. They expressed that although their view of psychiatry stems from biology they cannot sidestep the reality that pediatric psychiatry is highly influenced by political and cultural trends. They highlighted that a child’s functioning versus demands in their environment, to some extent builds the neuropsychiatric field, since there are no blood samples or other objective tests to verify ADHD. They felt pressure to prescribe medicine as politicians had stated that “all children should be able to pass their final exams”. (I3)

“Should one really prescribe medicine to a child that manages in school, will get grades in all subjects and will be able to get a good future, just because this child could be able to get an even better future?” (I8)

The clinicians underpinned the complexity of defining a child’s functioning to be above or below cutoff for an ADHD diagnosis, and that it could be risky to simplify the process. An example was given regarding a child presenting a clinical picture of ADHD, that later turned out to be suffering from a prolonged, complex crisis reaction.

“My experience is that trauma often is diagnosed later on, when a child diagnosed with ADHD is treated according to guidelines, but nothing helps, has had treatment for 2 years and is about to be discharged. Then it may turn out that this child has been sexually abused for years”. (18)

The clinicians highlighted the importance of being aware of the continuous development of a child’s functioning and skill acquisition, and how it unfolds over time. They reflected on the fact that a child’s expressed emotions were sometimes, in psychiatry, viewed as symptoms of ADHD, instead of seeing them as individual emotional responses to the surrounding circumstances. Clinicians asked themselves if ADHD perhaps can be seen as immaturity in relation to the environment’s demands and they referred to the fact that many adults no longer reach the criteria for an ADHD diagnosis.

#### 3.1.2. Encountering Parental Bias

The clinicians spoke of diagnosing ADHD as a first step towards deciding whether to prescribe medication or not. The clinicians experienced that it had become easier over time for parents to demand an ADHD assessment for their child. They stated that over the last decade such a request was almost always granted, leading to more children being assessed and consequently more being diagnosed.

A significant part of the information leading to diagnosis is the child’s narrated history, given by the parents or other caregivers, which was considered as challenging in itself. They speculated that parents have different intentions and skillsets that can affect their abilities to accurately describe their child’s symptoms and function. They experienced that parents and children self-diagnosed ADHD while waiting for an assessment.

“Parents that have been waiting for an assessment of their child for a long time, expect to get something back, they will not be so satisfied with, ‘No, it’s not ADHD’”. (I4)

As the clinicians often meet the children on just a few limited occasions, they found that parents have a great power to influence the child’s diagnosis. They reflected on the fact that parents might be biased when reporting symptoms based on the common knowledge on which information leads to a diagnosis and what would not.

“The most difficult thing with a diagnosis is that it’s the parents who present the function and medical history of their children when they were younger, and I can say that my experience is that a parent can say exactly the right things, so it sounds very much like ADHD. That is not so hard”. (I8)

#### 3.1.3. Holding Back Personal Opinions

The psychiatrists illuminated that each clinician is influenced by his or her own personal attitudes and experiences in the process of prescribing or not. They struggled with not letting their personal opinions influence their professional clinical decisions. As an example, they shared that when it came to their own children, they would be less likely to agree to ADHD medication compared to what they suggested to patients. On the other hand, it could be tempting to consider medication for their own children when they saw them struggling with homework.

They argued that to which extent a clinician explains the symptoms of a child to be biological or rooted in environmental and other factors, can eventually bias the decision to prescribe or not. Being more biologically oriented was considered to make the process fast and simple, to “just press play” and to prescribe medication.

The clinicians described that choosing to prescribe medication contrary to the unit´s culture/tradition, led to more work since they had to do all tasks themselves such as follow-ups. This curbed the desire of doctors to follow their own opinion, and instead made them to assimilate into the current unit culture.

“There are experienced, senior nurses who manage this independently and decide on increasing the dose in a way I would not have done, and that´s …, well, that’s where the local culture plays in”. (I9)

#### 3.1.4. Wanting the Child´s Best

The psychiatrists shared how they would try to tune in to the child, and his or her circumstances in school, when seeing the potential benefits of trying ADHD medication. They intuitively wanted to help and support the child and argued that it is often worth a try, to see if medication could support the child to better function in school and in getting better grades. These intentions made them prone to prescribe ADHD medication earlier in the treatment than suggested in guidelines.

“And one feels sorry for the child who sits there, attends secondary school and needs those grades, and then one thinks ’I will prescribe, let´s try ADHD-medication anyway’” (I5)

Frustration arose when clinicians recognized an evident ADHD diagnosis in a child waiting for assessment. They could not bypass the regional assessment waiting line, and they felt sorry for the child not being able to start with ADHD medication right away. They were eager to help the child immediately, so clinicians described that they might in such cases prescribe potentially effective off-label medications like bupropion which would not require a confirmed ADHD diagnosis.

“I prescribe the medication anyway as my goal is to make the child to be able to manage their homework or to be able to follow at least half of a lesson”. (I8)

#### 3.1.5. Persisting in Supporting Parents

The psychiatrists said that adaptations often are needed not only in school, but also in the home environment, and that the choice to prescribe today or to wait is made in the minute, when seeing the patient. They experienced that parental support varies to a significant degree. Some parents were considered to have capacities to adapt in such a way that a diagnosis is made too late, compared to some children receiving diagnosis prematurely due to parent’s lack of ability to support their child. The clinicians said that the longer time they had worked within CAMHS, the more they valued and placed emphasis on efforts to support parents in optimizing the child´s sleep, diet and physical activity, and to enhance family lifestyle and relations before prescribing medication.

“But I think in general we should be more open and honest, maybe not as strict, as for example regarding requirements for getting a gastric bypass surgery, that one first needs to lose many pounds, but a bit like it: that the family has to try out some things before ADHD medication is considered”. (I6)

The psychiatrists described that their unit previously had more focus on teaching parental strategies, but lately psychoeducation for parents had been cancelled due to lack of nurses. They wished to be able to see the families more often and at longer appointments to get to know them better. The clinicians described that the prescription of ADHD medication could correlate inversely with how well parents managed to put necessary support and adjustments in place. Although the clinicians talked to parents about the importance of relational and communicational adaptations, they experienced that they lacked tools and structure to help the families to successfully implement a change and to do follow-ups. Prescribing medicine before these changes were made was perceived as problematic.

“Families come after they have received psychoeducational interventions, but many of these families can have difficulties to implement the training in everyday life”. (I1)

### 3.2. The Culture at the Unit

The clinicians experienced they belonged to a local unit culture with a history and dynamics that influenced how ADHD medications were prescribed. They wished that the assessment of the child´s symptoms and function, prior to giving a diagnosis, was consistent with regional and national guidelines in every CAMHS. Such an alignment had not been implemented according to the clinicians.

#### 3.2.1. Noticing Regional Inconsistencies

The clinicians spoke about how they perceived that the four pediatric psychiatric clinics in the region worked in significantly different ways regarding both assessment and treatment of ADHD, and how they had been “raised” in that local tradition.

“It’s so obvious, if you are resident in /city X/ you are trained to follow the guidelines strictly, but if you instead got your training in /city Y/ you would probably work outside the book”. (10)

The psychiatrists argued that personal style is something that should be valued in clinical work, but one should at the same time be aware of the potential bias when it comes to how strictly one follows the guidelines. They sensed that the psychiatric field leaves more room to “play outside the book” compared to physical health care. Although following guidelines was seen as important, sometimes being symptom and function oriented was considered even more important.

They described variation between units concerning procedures created by the directors or the senior doctors. An example was about units with “fast track assessment” leading to quicker access to effective treatment for ADHD but a risk of neglecting co-morbidity. “Fast track assessment” was supposed to derive from wanting to work in a person-centered way and lead to sidestepping the guidelines that had created bottlenecks. They described units where senior doctors had created a culture of never prescribing medication for ADHD without a very thorough assessment, probably leading to high quality of care, but less children were able to get treatment and waiting lines tended to build up. Another variation experienced by the clinicians was with regard to the degree to which different units and school psychologists emphasized interventions such as trying out adjustments for the child or parental psychoeducation before assessment. They pointed out that some children were assessed too quickly and others, the obvious cases, should have been diagnosed much earlier.

“I would guess that if one would compare the different schools, those schools with psychologists delivering more individual interventions for their pupils would send less children with mild to moderate ADHD to CAMHS”. (15)

Another mentioned the inconsistent manner in which some regional units, contrary to guidelines, used ADHD medication like methylphenidate as a diagnostic tool. If the medication had an effect at follow-up, an ADHD diagnosis was set. As most people can perceive a positive initial effect with stimulants, the clinicians pointed to a risk of over-diagnosing when medication is used like this. As a contrast, putting the bar of assessment too high, was feared as it can lead to the creation of waiting lists of children in need of effective treatment.

The clinicians perceived a difference in the quality of the assessments between private and public units, stating that they believed that private units were more prone to diagnose ADHD compared to regional units. The clinicians described a challenge when needing to follow up the treatment plans and prescriptions, that other doctors had made, that differed from their own personal opinions. It was perceived as especially challenging when they met children with a very complex social background, who were often in foster families, treatment homes or state facilities. These children were often prescribed high doses of stimulants. The clinicians shared that it was important that they personally could see a substantial effect at follow-up, otherwise the medication should be stopped, like with all other medications.

“They come to my clinic with a diagnosis made at a private unit. Then we do have to prescribe stimulants, although the condition might be mild. I go through the assessment and think no, but I have to prescribe amphetamine anyway”. (11)

#### 3.2.2. Recognizing Cultural Dynamics

The psychiatrists had noticed that both ADHD assessment and treatment are more culturally dependent compared to other conditions in CAMHS, and that the threshold for diagnosis has been significantly lowered during the last decade. They discussed the involvement of the pharmaceutical industry and their campaigns of creating awareness of ADHD in society. They described that since evidence-based interventions and adaptations can only be implemented when a child is diagnosed, then they must diagnose. The clinicians wanted to challenge that positivistic mindset, that if clinicians find the right diagnosis, they find the right treatment.

“When I started in psychiatry much more was needed to get an ADHD diagnosis, and I believe that main reason for that change is related to the criterion regarding loss in everyday functioning”. (13)

They described how it is deeply rooted in the system, that you cannot prescribe ADHD medication without a diagnosis. The clinicians argued that reasons for them to hesitate when considering making an ADHD diagnosis can be both related to the fact that ADHD medication is classified as narcotic substance and traditionally has been perceived as potentially dangerous, and the fact that there is a demand for stimulants among young people which can see them being sold illegally on school grounds. On the other hand, they reflected on the fact that the time-consuming assessment protocols were not necessary in cases where ADHD was obvious. In their view, only very experienced psychiatrists had the confidence to prescribe the whole repertoire of available medications early in the process. This was perceived as a cultural phenomenon contrary to the issues regarding neuroleptics, which they considered to be prescribed with less hesitation as soon as a diagnose was set, despite known side-effects.

“I think they have created some kind of sacred aspect around ADHD medications considering that they are narcotics, classified as narcotics, and so on. And these sacred medications, you see them as something that, well you shouldn’t dare to prescribe them, but only in rare cases, and where everything is in place…” (10)

### 3.3. The Role of Society

The clinicians illuminated that prescribing patterns of ADHD medication stem from society at large. They pointed out that complex circumstances influence a child’s function and that ADHD diagnosis to some degree is dependent on their surroundings’ motivation and capacity to adapt to the present state of the child.

#### 3.3.1. Demanding too Much of the Child

The clinicians argued that the current school system, and sometimes parents, had high demands on the child which will have a significant impact on the child’s everyday function. This was exemplified by the statistical fact that “the later you are born in the school year, the more likely you will have an ADHD diagnosis” (I4). Today, they said, a child in the first years of school is asked to perform complex tasks independently. They shared their experience that these academic goals originate from children with high capacities of executive function, and they thought that 20–50% of children will not be able to reach it, no matter how pedagogics are twisted and turned. The clinicians said that this phenomenon of requiring too much of a child is not only true for the school system, but for society at large, where they noticed a general over-estimation of human capacity. They felt that when the society is “too medicalized, you forget to adapt the world to people”. (I4)

“And what does it say about our society and our school system that so many children require stimulants to be able to manage in school?!” (12)

The clinicians experienced that a discrepancy between a child’s capacity and the demands by the surrounding society, sometimes could be perceived as symptoms of ADHD. To prescribe ADHD-medication-like stimulants to children in order to momentarily compensate for the child´s lack of capacity and simply help them to pass their exams was seen as an ethical challenge. They wished instead that they could have supported these children in other ways to adapt and hoped that children with a lower level of academic performance will be more accepted by the society in the future.

“Sometimes you meet a family that has a 10-year-old who is not performing at his best. They have received an ADHD diagnosis and they have received good adjustments at school because they go to one of the slightly better schools. They have got excellent support, but the kid struggles a bit with math, for example. It´s not A-level and not B-level either, it’s C but “it’s my kid so I think he should be able to get an A”. (I3)

#### 3.3.2. Depending on Support at School

The clinicians experienced that adjustments at school to support a child’s function differed to a very large degree and were often dependent on the attitudes of the teachers, the principal and the school’s health-care team. Some schools, they said, do not even require an ADHD diagnosis to provide a support teacher, extra breaks or other support throughout the day. The clinicians felt that if a child has difficulties with concentration and regulating activity, this should initially be addressed and supported mainly by the school. However, they said, most children had to wait for optimal adjustments despite an ADHD diagnosis, sometimes leading to lowered thresholds to prescribe ADHD-medication prematurely. From this point of view, it was argued that variation of prescription of ADHD medication starts in school, not at CAMHS.

“For me as a doctor it’s an ethical dilemma having a patient with ADHD symptoms when you as a doctor think that most of all, adjustments in school would be beneficial”. (14)

The clinicians described how they sometimes diagnosed a mild form of ADHD and prescribed medication immediately if the school had a reputation of insufficient capacity to implement necessary adjustments and support. After having sent documentation about adjustments needed and after having attended lots of school meetings in vain, they prescribed ADHD medication to compensate for an inadequate school environment.

“In the end, it’s me sitting there somehow trying to treat imbalances in the school system”. (13)

#### 3.3.3. Lacking Time and Trust in Colleagues

The psychiatrists experienced a lack of time to get to know the families. They pointed to the fact that their own, and their unit’s, resources were not modified to suit the ever-increasing requests for ADHD assessments. They also received many patients from private clinics that had diagnosed the child with ADHD and that now there was an expectation for them to prescribe medication to children that they had not assessed themselves, without much time to read their medical history. Likewise, the psychiatrists had difficulties in trusting the assessments made by the nurses who met the families regularly and said that they would prescribe less medication if they themselves had time to talk to the parents and if they would be able to do more assessments themselves.

“Much today is assessed via the nurse, and thereby in several steps. Parents call and tell the nurse things, the nurse makes her own assessment, then we get a short summary in a note and we then often decide to increase the dose. Was it really necessary to increase that dose is the question?” (I1)

This led to a strange feeling of not applying “safety principles” when prescribing ADHD medication. The psychiatrists had a significantly increased workload and could not carry out assessments and follow up treatment in the best way. Lack of time was considered to increase the prescribing of ADHD medication.

#### 3.3.4. Inheriting Socioeconomic Challenges

The clinicians spoke about how children are born into a family system without possibility to choose. Parental economics and geographical location were perceived to influence how much a child’s ADHD symptoms would become a problem or not. They described how children that had a long way to travel to CAMHS, with single mothers and several siblings at home, often with their own psychiatric conditions, just could not take part in the non-pharmaceutical interventions offered. They reflected on the fact that the socio-economic load on a child can be multifactorial and will always contribute to care being unequal, and to the extent a family might be able to implement things they learnt during psychoeducation in their everyday life. When the family´s capacity was judged to be very limited the clinicians prescribed medication earlier in the treatment process.

“In our most burdened municipalities where there are lower levels of education, I think that psychiatrists are more inclined to medicate the children”. (I5)

## 4. Discussion

This study is, to our knowledge, the first to investigate attitudes and experiences of diagnosing and prescribing medication for ADHD among pediatric psychiatrists in Sweden. The results demonstrate that this part of pediatric psychiatry is surrounded by several areas of concern.

The participants described uncertainty and a lack of confidence when meeting children and their families who were seeking care for symptoms they suspected to be ADHD. The psychiatrists were not as convinced as the families were that medication in all cases was the right intervention but experienced pressure, both by parents and other professionals, to prescribe pharmacological treatment. Although all participants were employed by the same organization and are supposed to follow the same protocols regarding assessment, treatment and follow-up, individual tendencies influenced the decision-making whether to prescribe stimulants or not and were fueled by culture and traditions at the local outpatient unit the consultant was working, and by the influence of society at large. This phenomena of inter-variability among psychiatrists’ assessments has been recognized and studied before, being a significant motivation for the development of guidelines, such as DSM, that have emerged to counteract this unreliability [24]. Further developments in the DSM-IV and DSM-V have slightly increased psychiatrists’ diagnostic consistency, but still major decision heterogenicity is found [25].

Diverging attitudes towards the diagnosis and treatment of ADHD has previously been shown among general practitioners in several countries [26]. In settings where the practitioners are gate keepers, their personal attitude influences how children with suspected ADHD are referred to specialists, leading to regional differences concerning prevalence rates of ADHD. In Germany, pediatricians may diagnose and treat ADHD. A survey, with German pediatricians, reported that they assessed ADHD and treated with multimodal therapy in line with the medical guidelines [27]. However, Germany still has regional differences, like Sweden [22]. This raises questions about individual doctors’ obligations to adhere to national guidelines versus their ability to divert from protocol to allow for individual decisions based on their professional beliefs. How far can we accept individual attitudes and opinions to guide decisions on care without letting go of the demand for equal care? The pediatric psychiatrists in the present study declared an intention to strive for the child’s best, in line with their Hippocratic oath and were prone to accept deviations from evidence-based guidelines for the best of the child. At the same time, the psychiatrists were acknowledging a family’s special needs.

The recently released guidelines for ADHD from The Swedish National Board of Health and Welfare [28] state that diagnostic assessments for ADHD should be performed by a team consisting of at least one doctor and one psychologist, and regarding treatment that ADHD patients should be offered multimodal treatment, including pharmacological treatment if it is considered appropriate. Haynes et al. state that clinical expertise is based not only on efficacy, effectiveness, and the efficiency of available treatment options, but also on the patient´s preferences and what interventions she or he will accept [29]. The physician needs to consider these factors and recommend the treatment that the patient will agree on and follow [29].

On the other hand, the Swedish Patient Act emphasizes that all children should receive equal care, regardless of geographical and socioeconomic factors and that all patients should obtain information on all evidence-based treatment options [30]. Given the diverse nature of ADHD, it could be argued that some children could benefit from individualized therapeutic approaches although presenting similar symptoms. Current guidelines recommend stepwise care in order to maximize benefit and avoid adverse effects. If, according to evidence and clinical experience, more than one option exists, the patient should be offered the option of the preferred alternative. If psychiatrists interpret guidelines differently, or comply to guidelines in varying degrees, not every child might be offered all treatment options, and the ideal of equal health-care is threatened.

The clinicians acknowledged the rationale behind using protocols, while at the same time pointing to local units not necessarily adhering to them. They shared their experience of CAHMS units having different working cultures, which influenced their way of structuring their work and even nudged their decisions in diagnosing and prescribing related to ADHD. The clinicians highlighted how senior doctors and unit directors significantly influenced the variance between units. Newly employed doctors experienced a need to align with the local unit variation for pragmatic reasons, if not, the workload on them increased. That health care units have their own unique working climate has been demonstrated in previous research and factors such as good balance between independence, engagement, loyalty and acceptance seems to be important for successful units in regards of implementation [31].

Several possible reasons for asymmetric adherence to guidelines, at the different units, were mentioned by the consultants including the personal style, experience and training of the clinician, the surrounding environment, both at the unit and in society at large. This has previously been suggested by Mykletun et al., who currently are studying these factors among clinicians in Norway [17].

According to the interviewed clinicians, one of the reasons behind non-compliance to protocol, both individually and at the unit, is the wish to be autonomous and to be in charge of the assessment, as trusting in someone else’s judgment, someone that they do not know that well, or even disagree with is perceived as challenging. A reluctance from general practitioners to collaborate with specialists about children with ADHD have been shown in earlier studies [26]. The clinicians in the present study revealed a mistrust of colleagues in private clinics who diagnosed children, who were then referred to the regional units for pharmacological treatment. This raises questions on how well a diagnosis made by another doctor or by a psychologist is accepted. Whether private clinics diagnose children with ADHD in a higher or lower degree compared to regional clinics has yet to be studied.

The informants also revealed a mistrust regarding nurses’ assessments, despite them having regular contact, over time, with the families, to among other things follow up the effectiveness of prescribed medicines. As teamwork is a core competence, acknowledged by the national assembly of medical doctors and nurses [32], and by the national guidelines [10], it is surprising that participants preferred to work on their own. To allow for successful teamwork in the future these areas should guide future discussions about organization and teamwork at the units.

Different ways of organizing care for children seeking help for ADHD have their unique consequences, as stated by the consultants. The “fast track assessment” unit culture potentially enables evidence-based care earlier and indeed to more patients than a more “slow assessment unit”. On the other hand, it might increase the risk of over inclusion, over treatment and missing co-morbidity. According to the recent Swedish guidelines from the national board of health and welfare early interventions should be put in place before the patient has been assessed for a diagnosis [28]. Patients with current problems suggestive of ADHD should get support in their school and at home even if they do not fulfill the criteria for a clinical diagnosis [10].

The pediatric psychiatrists in the present study underlined their experience of interdependence with the school. According to them, one of the reasons for the variation in ADHD assessment and treatment is rooted in variation in how schools take care of children with special needs. This finding is not new and have been demonstrated before to be an underlying factor in ADHD assessment and treatment [28]. Based on this observation, it could be argued that continued and even enhanced attention should be directed to support the schools in their work with children with ADHD symptoms.

To agree on a common definition of “normal” function of a child and, deriving from that, what to expect and require from a child was recognized as an area of concern by the pediatric psychiatrists in the present study. This topic is not new. As the limits between ADHD and normality are diffuse, not only the child but also the psycho-social environment should be examined [33]. It could be argued that schools today in Sweden have never been as enlightened regarding individual asymmetries in cognitive skills and implementations of evidence-based adaptations to buffer it. Still, the clinicians in the present study commonly experienced meeting children they believe still are “under too much demand” and therefore creating, or/and increasing, ADHD symptomology.

### Strengths and Limitations

The between-country and within-country variations in ADHD diagnosis and medication rates are large and have been described previously [17]. Yet, the qualitative shades behind it expressed by the pediatric psychiatrists in the present study are novel, adding important perspectives to better understand differences in diagnostics and therapy [27]. Their reflections around the decision making on prescribing or not, deepens the understanding of processes around these choices. The complexity in the diagnostic process, with varying comorbidities and varying psycho-social circumstances, may lead to difficulties following the guidelines and may cause inter subjective variability, but this can hardly explain the huge differences in ADHD diagnosis and medication between geographical areas. The wide range of years of clinical experience as a specialist, equal distribution of gender and inclusion of clinicians in both outpatient and day care units among the participants is a strength of this study. All 36 pediatric psychiatrists employed by Region Skåne were invited, and thus given the opportunity to share their experiences. Eleven of them participated which is a fairly high proportion. Due to the pandemic COVID-19 the interviews were conducted online via Zoom possibly contributing to a distance between the interviewer and the interviewee. However, the informants spoke openly, giving rich data. The qualitative study design with structured analysis, conducted by two of the authors, allowed any attitude or experience to be recognized, including opposite and diverging experiences in the result. In the next step the results were discussed with the other authors. The authors are psychiatrists and psychiatric nurses with long experience, adding different professional perspectives to the interpretation and discussion. Our pre-understandings were reflected on. It is not known if psychiatrists with strong opinions about the topic were more prone to take part in the study. Qualitative research does not produce repeatable results, and to which degree the experiences of the participants in the present study are transferable is unknown.

Only psychiatrists employed in public health care within the region were invited to the study. Since psychiatrists employed at private clinics assess many children with ADHD symptomology it would be interesting to illuminate their experiences as well. None of the pediatric psychiatrists at the in-patient unit participated, which is a limitation.

## 5. Conclusions

In this study, the participating pediatric psychiatrists described how they balanced clinical guidelines with personal attitudes and demands from the patient’s family, different local unit cultures, and society at large in a complex interplay. The study adds information about how attitudes may lead to variation in diagnostics and therapy in children with ADHD. In order to increase adherence to evidence-based medicine and promote equal health-care utilization, these findings indicate that collaboration with schools and other caregivers, as well as promoting teamwork, is of particular importance and should be the focus of future research.

## Figures and Tables

**Table 1 ijerph-20-00221-t001:** Themes and subthemes.

Meeting the Family	The Culture at the Unit	The Role of Society
Defining normal function	Noticing regional inconsistencies	Demanding too much of the child
Encountering parental bias	Recognizing cultural dynamics	Depending on school agency
Holding back personal opinions		Lacking time and trust in colleagues
Wanting the child´s best		Inheriting social inequality
Persisting in supporting parents		

## Data Availability

The data used in this study contains sensitive information about the study participants and they did not provide consent for public data sharing. The current approval by the Regional Ethical Review Board in Lund, Sweden (Ethical Review Board no. 2021-01563) does not include data sharing. A minimal data set could be shared by request from a qualified academic investigator for the sole purpose of replicating the present study, provided the data transfer is in agreement with EU legislation on the general data protection regulation and approval by the Swedish Ethical Review Authority.

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
