# Peer review of "The Winding Road to Equal Care: Attitudes and Experiences of Prescribing ADHD Medication among Pediatric Psychiatrists: A Qualitative Study"

_ijerph, 2022, doi:10.3390/ijerph20010221_

Round 1
Reviewer 1 Report
There is an ongoing debate regarding the causes of increased ADHD diagnosis worldwide. Furthermore, drug prescription for ADHD pediatric patients is also a controversial issue. Authors mention that ADHD incidence, as well as drug prescription for ADHD patients is different among the 4 different clinics in the Skane region, finding less prescriptions of pharmacological treatment in those clinics with less ADHD diagnosed patients. Author’s aim to determine the reasons for the variation in diagnosis and prescription patterns in the different clinics.
The article reviews the opinion of 11 pediatric psychiatrists from different clinics in Skane regarding the reasons leading them to diagnose a child with ADHD, as well as the reasons supporting their decision to prescribe pharmacological treatment for these children. The issues addressed are quite relevant given the worldwide increase in ADHD diagnosis, as well as the increased use of pharmacological treatment, which is generally effective but also has a number of drawbacks.
The results of the study reveal a rather large number of factors influencing both ADHD diagnosis and the prescription of pharmacological treatment, far beyond the mere presence of ADHD symptoms. For example, psychiatrists prescribe pharmacological treatment for a number of reasons, including their opinion regarding the biological nature of the symptoms, the historic tendency of a particular health unit to prescribe pharmacological treatment or the follow up work implied in the decision to prescribe it, as well as their view regarding possible academic improvements for the children if medicated, among others. Similarly, several factors influence ADHD diagnosis, including parent´s description of child´s behavior, the tendency of specific health units towards a fast track assessment and, in some regional units, even the use of methylphenidate as a diagnostic tool, prescribing the drug and diagnosing ADHD when symptoms decrease at follow up. In all cases, a mixture of clinician´s and parent´s perspective, as well as school and society demands for children´s performance, play an important role in ADHD diagnosis and drug prescription for these patients.
As mentioned by authors, the number of children diagnosed with ADHD has increased, as well as the prescription of pharmacological treatment for these children. These two factors have raised public attention due to several factors including, as mentioned by the authors, whether the condition is over diagnosed and if pharmacological treatment is the best choice to treat these children. The concerns are strengthened by the fact that not all clinicians diagnose and treat ADHD in the same way, suggesting that it is possible that some of the children might not present ADHD, or if diagnosed with ADHD might be treated with a non-pharmacological approach if assessed by a different clinician, thus the relevance of determining the factors influencing decision making in health professionals.
While the number of subjects interviewed is certainly small and they are from a specific region, as authors mention when addressing the limitations of their study, the results are nevertheless interesting given the detail of the analysis and the relevance of the question.
I only suggest two additions to the present manuscript:
1.- In the section 2.1 (in methods), authors mention the study design and provide a reference where this approach is described. To strengthen their conclusions, it is advisable to include in this section a brief description of the main features and advantages of this type of analysis.
2.- There are many factors associated with ADHD occurrence, and pharmacological treatment does not have the same effect on all patients (including lack of response and a different degree of adverse reactions). In addition to this, authors describe the influence of many factors on both ADHD diagnosis and treatment selection, yielding a complex scenario in regard with this disorder and its treatment. Given the diverse nature of ADHD origin, would it be possible that children under different circumstances, with an ADHD possible originated by different factors, would benefit from a different therapeutic approach in spite of presenting similar symptoms? Or would it be more beneficial for patients if clinicians followed rigorously the local guidelines in all cases in spite of their personal opinion regarding each children? Please comment in the discussion section
Author Response
Authors Reply to review 1, all authors have contributed and agreed on the revisions made
Comment 1: In the section 2.1 (in methods), authors mention the study design and provide a reference where this approach is described. To strengthen their conclusions, it is advisable to include in this section a brief description of the main features and advantages of this type of analysis.
Author's Reply: the authors do agree on the benefit of such an addition, and we have added a brief description of both features and the advantages of qualitative analysis in the method section (section 2.1 design: line 88c; section 2.5 data analysis, line 120; and section strengths and limitations, line 634).
Comment 2: There are many factors associated with ADHD occurrence, and pharmacological treatment does not have the same effect on all patients (including lack of response and a different degree of adverse reactions). In addition to this, authors describe the influence of many factors on both ADHD diagnosis and treatment selection, yielding a complex scenario in regard with this disorder and its treatment. Given the diverse nature of ADHD origin, would it be possible that children under different circumstances, with an ADHD possible originated by different factors, would benefit from a different therapeutic approach in spite of presenting similar symptoms? Or would it be more beneficial for patients if clinicians followed rigorously the local guidelines in all cases in spite of their personal opinion regarding each children? Please comment in the discussion section.
Author's Reply: The authors agree on the fact that this area deserves more focus in the manuscript, and we have added comments regarding this topic both in the discussion (from line 540) as well as in strengths and limitations (from line 623.
Reviewer 2 Report
Authors have explained an important concern about the inner workings of a system and the variability that occurs with children and families. Authors may want to consider highlighting the complexity of diagnoses and the ethical considerations a Pediatric Psychiatrist may face in following multimodal guidelines. This inter subjective variability is complex but will be alleviated by discussing it in the limitations. This may increase the reader's viewing to a broader audience. Furthermore, highlighting the motivation for writing this article is unclear. Is it to work with a system or is it to target a certain subgroup.
Author Response
Response to review 2, all authors have contributed and agreed on the revisions made
Comment 1: English language and style are fine/minor spell check required
Author’s Reply: Thank you for highlighting this; a new, final language editing has been done, and the few misspellings found are now amended.
Comment 2: Are all the cited references relevant to the research? Can be improved
Author’s Reply: Thank you. We have looked through the references and do agree on the fact that the previous reference 32 (on teamwork, by the Swedish nurse and medical association) should be replaced by an international reference on that topic, now added to the manuscript. Regarding the other cited references, we want to argue that they are both relevant but also necessary to include.
Comment 3: Is the research design appropriate? Can be improved
Author’s Reply: Thank you. We have now further clarified our rational for choosing a qualitative research design in the methods section, under study design (line 88, and line 120).
Comment 4: Are the methods adequately described? Can be improved
Author’s Reply: Thank you for this suggestion, please find our revised version of the materials and methods section (from line 88 and onwards), in line with our reference to the method used (Lindgren et al, 2020).
Comment 5: Are the results clearly presented?? Can be improved
Author’s Reply: Thanks for this comment. The authors have been discussion alternative ways of presenting our results but see challenges with that. The way the results are presented in the article are in line with the method by Lindgren et al 2020, and in line with previously published qualitative papers, a presentation that highlights each quote chosen. We would suggest keeping the presentation of the results as it is.
Comment 6: Are the conclusions supported by the results? Can be improved
Author’s Reply: Thank you for highlighting this issue. We have added content regarding further research to improve the conclusion section (line 658). With that addition we would argue the conclusions are supported by the results.
Comment 6: Authors may want to consider highlighting the complexity of diagnoses and the ethical considerations a Pediatric Psychiatrist may face in following multimodal guidelines. This inter subjective variability is complex but will be alleviated by discussing it in the limitations. This may increase the reader's viewing to a broader audience
Author’s Reply: Very good suggestion. We have revised the manuscript according to that by adding content to the strengths and limitations section (from line 623).
Comment 6: Furthermore, highlighting the motivation for writing this article is unclear. Is it to work with a system or is it to target a certain subgroup.
Author’s Reply: Thank you for commenting on this. We have clarified that topic by adding content in the introduction (from line 81) but also by changing some wordings in the introduction. Taken together we think that the reader will understand better now that our motivation for writing this article is to support and promote equal health care utilisation/to work with a system.